# OpenReview forum: "AdaGC: Enhancing LLM Pretraining Stability via Adaptive Gradient Clipping"
_ICML.cc/2026/Conference — ICML 2026 regular_

### Official Review · Reviewer_vE7D · 2026-03-09

**Soundness:** 3
**Presentation:** 3
**Significance:** 2
**Originality:** 3
**Overall Recommendation:** 5
**Confidence:** 3

**Summary:**

The paper proposes AdaGC - a practical gradient clipping method intended to reduce the appearance of loss spikes. Loss spikes is a prominent problem that harms the training stability of LLMs and can be caused by various triggers ranging from data quality to hardware faults. Common mitigation approaches include data filtering, skipping batches, careful hyperparameter tuning, etc. Successful mitigation often require to identify the exact cause of the loss spike to select the proper approach. The authors note that despite the variety of the upstream causes, they always result in the abnormal gradients, so the proposed solution focuses on fixing the gradient problem. The core idea of the proposed method is to bound per-tensor gradient norms relative to the EMA of the tensors historical clipping values. It is a simple approach with minimal overhead and high applicability that is agnostic to the exact loss spike cause. The experiments show that the method successfully mitigated loss spikes and improves the downstream performance over a several baselines.

**Compliance With Llm Reviewing Policy:**

Affirmed.

**Final Justification:**

Authors addressed my concerns during the rebuttal. I believe this paper a provides valuable practical guidance for reducing loss spikes.

**Key Questions For Authors:**

1. Have you tested Muon and Lion without any gradient clipping? If yes, does it show the same stable performance? Have you tried to induce the spikes, for example, by feeding the model with garbage data?

2. Why not including Table 12 results in the main paper? These baselines are more relevant and would strengthen the claim.

**Limitations:**

The paper doesn't discuss limitations.

**Strengths And Weaknesses:**

## Strengths

1. The proposed method is simple and efficient. It is optimizer-agnostic and easy to integrate into training with just minimal memory overhead for EMA values.
2. The paper is clear and presents the problem well. The paper discusses the various causes of the loss spikes and observes that they all result in the same abnormal gradient behavior which justifies the motivation to develop a gradient-focus solution.
3. Extensive experiments over three models demonstrate that AdaGC not only mitigates the spikes, but also improves the models performance across several benchmarks.
4. Since the method only relies on the local tensor information it is applicable in distributed training settings by removing the need for global information aggregation.

## Weaknesses

1. Experiments in the main paper only include simple baselines like GlobalGC and AGC, while more relevant approaches that aim to reduce loss spikes, like WeSaR and SPAM, are only compared to in Table 12 in appendix. Notably, although AdaGC shows the performance improvements over most of the tested baselines, the gain remains marginal, especially on smaller model.
2. The method introduces additional hyperparameters for EMA: $\beta$, $\lambda_{rel}$ and warmup steps $T_{start}$. Although the authors claim that the performance remains stable across different values, Tables 3 and 4 show that the model performance is sensitive to hyperparameter choice and requires an extensive grid search. Figure 6 also shows a significant difference in convergence for different $T_{start}$ choices. Moreover, the hyperparameter search is done on Llama-2 7B model only and it is not clear whether the same choice is optimal for different models.
3. Results on Muon and Lion are not convincing because the baseline versions of both optimizers are already stable and don't show any loss spikes in the tested setting.

---

> ### Author Rebuttal · Authors · 2026-03-30
>
> We thank the reviewer for the constructive feedback and will address each concern.
>
> **W1: SPAM and WeSaR only in appendix**
>
> As described in Section 5.1 (Comparison Methods), we organized baselines by method category: clip-based methods (GlobalGC, AGC) are placed in the main paper as direct same-category comparisons, while non-clip methods (SPAM, WeSaR) are thoroughly evaluated in Appendix Tables 12-13. The appendix results actually favor AdaGC:
>
> - **Llama-2 1.3B**: AdaGC 46.33 vs SPAM 45.58 (+0.75%), both with 0 spike score
> - **Llama-2 7B**: AdaGC 51.01 vs SPAM 48.85 (+2.16%), SPAM still has 3 spikes
> - WeSaR eliminates spikes but requires special initialization (only for training from scratch), while AdaGC works for both from-scratch and resumed training
>
> We will add a brief summary of these results in the main text with a reference to the appendix.
>
> **W2: Hyperparameter sensitivity**
>
> Through grid search and cross-model validation, we recommend **λ_rel=1.04, β=0.99** as default values. These defaults work well across all tested models without re-tuning: Llama-2 7B, Mixtral 8×1B, ERNIE 10B-A1.4B, and **Qwen3 1.7B** in new experiments (see response to Reviewer vxi5 for details). This cross-model portability demonstrates that practitioners can use our defaults directly without extensive hyperparameter search.
>
> **W3: Muon/Lion experiments less convincing**
>
> This is a fair observation. The Muon and Lion experiments were designed to verify **optimizer-agnostic compatibility**—i.e., that AdaGC does not interfere with normal training or introduce side effects. We should have stated this purpose more clearly.
>
> **Key Question: Muon/Lion without clipping?**
>
> We tested Muon **without any gradient clipping** and observed stable training without loss spikes in our small-scale experiments. This aligns with Muon's spectral normalization design: it normalizes gradients and uses `msign` to extract gradient direction, making it inherently robust to gradient magnitude variations.
>
> However, **at scale, Muon still requires clipping mechanisms**. The recent KIMI K2 technical report (arXiv:2507.20534) documents this finding:
>
> > "Scaling up Muon training reveals a challenge: training instability due to exploding attention logits, an issue that occurs more frequently with Muon but less with AdamW... Max logits at this level usually result in instability during training, including significant loss spikes and occasional divergence."
>
> KIMI K2 proposed **MuonClip** (Muon + QK-Clip weight clipping) to address this, training 15.5T tokens with zero loss spike. This demonstrates that even optimizers with built-in normalization like Muon benefit from additional clipping mechanisms at scale.
>
> This finding reinforces our position: different optimizers exhibit different stability characteristics, and AdaGC provides a general-purpose gradient-level defense across all of them.

---

> > ### Author Rebuttal · Reviewer_vE7D · 2026-04-01
> >
> > Thank you for the detailed answer. I'll increase my score.

---

### Official Review · Reviewer_Z3cz · 2026-03-11

**Soundness:** 3
**Presentation:** 3
**Significance:** 3
**Originality:** 2
**Overall Recommendation:** 5
**Confidence:** 3

**Summary:**

The paper studies loss spikes in LLM pretraining and argues that, although the triggers are diverse, they often end up as abnormal gradients that contaminate optimizer states.
To address this, the authors propose AdaGC, an adaptive per-tensor gradient clipping method. Instead of one global threshold, each tensor gets its own threshold based on an EMA of its past gradient norms.
Experiments on Llama-2 7B, Mixtral 8×1B, and ERNIE 10B-A1.4B show that AdaGC reduces the reported spike score to zero and often improves downstream accuracy over GlobalGC.

**Compliance With Llm Reviewing Policy:**

Affirmed.

**Final Justification:**

Good paper and strong experiments. My concerns are fully resolved. I keep my score of 5.

**Key Questions For Authors:**

- Could you show me the results for Figure 2 in longer horizon? In my view, many quantities can be stable under a constant LR with enough iterations. So the oscillation of second moment norm is weird to me. If this is actually the case here, could you give me some explanation?
- In my understanding, the algorithm uses global GC in warm-up stage. In Figure 6(a), a global GC exhibits spikes many times before 20k steps. However, in Figure 6(b), a T_start=20k never have loss spikes before 20k steps, which is out of my expectation. Could you clarify about this?

**Limitations:**

There is no limitation section. I hope the authors will discuss these limitations based on the weaknesses I have mentioned and their own perspectives.

**Strengths And Weaknesses:**

Strengths:
- First of all, I think the writing is good. The introduction section is logical, clear, and informative.
- The method is simple, which is good, and the experiments are extensive, including both dense and MoE models, and also with different optimizers like AdamW, Muon, and Lion. Also, the experiment scale goes up to 10B-A1.4B and 1T tokens.
- The ablation study is good. The paper checks EMA initialization, warm-up length, adaptivity, and locality, which helps support the design choices.

Some weaknesses:
- Are there any failure cases for the AdaGC algorithm? Is this the end of studying training stability? I hope to see the authors' comments.
- What kind of instability sources can be mainly solved by the AdaGC algorithm and which cannot? While the experiments in Figure 1 classified by causes of instability is good. The main experiments seem not to follow this style. The only one tested is the choice of optimizer hyperparameters.

---

> ### Author Rebuttal · Authors · 2026-03-30
>
> We thank the reviewer for the positive evaluation and thoughtful questions.
>
> **Q1: Figure 2 results in longer horizon**
>
> We have provided extended training curves in **Appendix Figures 10-15**, showing the same quantities over longer training horizons. We encourage the reviewer to examine these figures. Additionally, even if optimizer states eventually stabilize after a spike, the transient contamination has already caused irreversible damage—the model is pushed to a worse loss basin, as confirmed by the downstream accuracy differences in Tables 5-6.
>
> **Q2: Figure 6(a) vs 6(b) — apparent contradiction**
>
> Thank you for this careful observation. We believe there is a misunderstanding about the T_start values:
>
> - **Figure 6(a)**: GlobalGC is used **throughout the entire training** → spikes appear before 20K steps.
> - **Figure 6(b)**: This is a T_start ablation. The **maximum warmup tested is T_start=2K steps** (not 20K). After warmup ends, AdaGC immediately takes over → **no spikes after 2K steps**.
>
> This is not a contradiction but rather a demonstration of AdaGC's effectiveness: once AdaGC takes over (as early as step 2K), all subsequent spikes are eliminated, contrasting sharply with the GlobalGC-only baseline in Figure 6(a).
>
> **W1 & W2: Failure cases and limitations**
>
> AdaGC is certainly not the end of studying training stability. As stated in our paper, AdaGC provides defense at the optimizer level without investigating root causes.
>
> **Concrete failure case**: We reproduced the loss spikes in GPT-2 training caused by low-precision Flash Attention, as described in the ICLR 2026 paper "Why Low-Precision Transformer Training Fails: An Analysis on Flash Attention". AdaGC was unable to eliminate these spikes. This may be related to engineering implementation details of Flash Attention (see also: Flash Attention GitHub repository, Discussion #1931 by Tri Dao et al.).
>
> In summary, AdaGC is most effective against **sudden gradient anomalies** from diverse causes (data corruption, hardware faults, optimizer hyperparameter sensitivity, numerical precision issues). For instabilities arising from other mechanisms, complementary approaches remain necessary. We will add a **Limitations section** discussing these boundaries.

---

> > ### Author Rebuttal · Reviewer_Z3cz · 2026-04-02
> >
> > Thank you for the rebuttal. I will keep my score, and I would like to continue the discussion with the authors.
> >
> > First, it is interesting to see that AdaGC fails in some cases. For research on training stability from the optimizer perspective, however, I think an important goal is to distinguish between instability issues that can be resolved by the optimization algorithm itself and those that cannot. What is your view on this distinction? More concretely, how should we identify problems that are feasibly solvable by optimizer design?
> >
> > Second, in the paper, the authors list four sources of instability: data quality, hardware faults, optimizer hyperparameters, and numerical precision issues. The first two are easy to understand. For the latter two, I want to confirm my understanding. For numerical precision issues, do you specifically mean using lower-precision RMSNorm? And for hardware faults, do you mean stochasticity in FlashAttention backward passes?
> >
> > Also, I hope the authors can add one limitations section in the paper.

---

> > > ### Author Response · Authors · 2026-04-02
> > >
> > > Thank you for the continued discussion. These are excellent questions that help
> > >   sharpen the contribution of our work.
> > >
> > >   **Q1: Distinguishing optimizer-solvable vs. unsolvable instabilities**
> > >
> > >   We propose the following framework:
> > >
> > >   **What AdaGC CAN solve — sporadic gradient anomalies.** AdaGC is effective when
> > >   instability manifests as *statistical outliers relative to the historical
> > >   gradient distribution* — discrete, sporadic events within otherwise normal
> > >   training. Examples include data corruption, hardware faults (e.g., GPU memory
> > >   bit flips), optimizer hyperparameter sensitivity (e.g., small ε amplifying
> > >   gradient spikes, as in our Table 6), and certain numerical precision issues.
> > >   The key criterion is: if the abnormal gradient is *detectable as an outlier*
> > >   by comparing against the EMA-tracked history, AdaGC can clip it.
> > >
> > >   **What AdaGC CANNOT solve — we identify two categories:**
> > >
> > >   *(1) Cumulative numerical errors from algorithmic engineering implementations.*
> > >   Our concrete failure case (low-precision Flash Attention) falls into this
> > >   category. The root cause is not sporadic gradient spikes, but rather small
> > >   numerical approximation errors introduced at every training step by the
> > >   engineering implementation of the algorithm. These errors accumulate gradually,
> > >   causing the gradient distribution to drift slowly. Since AdaGC's EMA adapts to
> > >   the evolving distribution, it cannot distinguish this drift from normal training
> > >   dynamics — there is no "outlier" to detect.
> > >
> > >   *(2) Silent bugs in training infrastructure.* Real-world training suffers from
> > >   diverse silent bugs — logic errors in framework code, misconfigurations,
> > >   communication data loss, etc. — where the training "correctly executes incorrect
> > >   logic." Gradients may appear statistically normal (no spikes), yet the training
> > >   outcome is wrong. This is fundamentally beyond what any optimizer-level mechanism
> > >   can detect. We refer the reviewer to Jiang et al. (OSDI 2025, "Training with
> > >   Confidence", arXiv:2506.14813, Table 3) for a systematic taxonomy of such
> > >   silent training errors.
> > >
> > >   **In summary**, the distinguishing criterion is: *Does the problem manifest as
> > >   detectable gradient-level anomalies?* If yes, AdaGC can help. If the problem is
> > >   systematic (cumulative numerical drift) or invisible at the gradient level
> > >   (silent code bugs), other approaches are necessary.
> > >
> > >   **Q2: Clarification on instability sources**
> > >
> > >   Thank you for checking your understanding.
> > >
> > >   - **Numerical precision issues**: This refers to the numerical precision used
> > >     in LLM training (FP32/BF16/FP16/FP8). Your understanding is partially
> > >     correct — low-precision RMSNorm is indeed one example. In our paper, we
> > >     present two concrete cases: (1) RMSNorm computed in BF16 vs. FP32 produces
> > >     different numerical results, with BF16 being less stable. Modern
> > >     implementations (e.g., Qwen3_5RMSNorm in Hugging Face Transformers) cast
> > >     inputs and weights to FP32 before computing RMSNorm to improve training
> > >     stability. (2) DeepSeek's FP8 blockwise quantization, which mitigates
> > >     the impact of outliers to some extent, thereby improving training stability.
> > >     More broadly, any precision-related source that causes gradient anomalies
> > >     falls into this category.
> > >
> > >   - **Hardware faults**: This refers to actual hardware failures such as GPU
> > >     memory bit flips and NVLink/InfiniBand communication errors — not the
> > >     stochasticity in Flash Attention backward passes. Flash Attention's
> > >     non-determinism is an engineering implementation choice for high throughput,
> > >     arising from the non-deterministic accumulation order of dQ, dK, dV in the
> > >     backward pass, which is a separate concern from hardware faults.
> > >
> > >   **Q3: Limitations section**
> > >
> > >   We will add a Limitations section covering: (1) the boundary of optimizer-level
> > >   solutions as discussed above; (2) the concrete failure case (low-precision Flash
> > >   Attention); and (3) the complementary relationship with architectural fixes,
> > >   correctness checking, and numerical analysis approaches.

---

### Official Review · Reviewer_TMiS · 2026-03-12

**Soundness:** 3
**Presentation:** 3
**Significance:** 3
**Originality:** 3
**Overall Recommendation:** 5
**Confidence:** 3

**Summary:**

This paper proposes AdaGC, an adaptive per-tensor gradient clipping scheme that mitigates such contamination by bounding gradient norms relative to a tensor-wise exponential moving average of their historical clipped values. AdaGC is optimizer-agnostic, with minimal memory consumption and communication overhead, and good compatibility to hybrid parallelism. Experiments show that AdaGC could eliminates training instabilities, and reduce loss spikes.

**Compliance With Llm Reviewing Policy:**

Affirmed.

**Final Justification:**

The convergence guarantee of Adam + AdaGC is already available.
Although there is no theoretical analysis of how to tune the hyperparameters, the Practitioner's Guide is also acceptable.
In overall I think this is a good paper. Thus I keep the positive score.

**Key Questions For Authors:**

1. Is there any corresponding theoretical analysis of convergence for AdaGC + SGD?
2. Is there any corresponding theoretical analysis of how to choose $\lambda$ and $\beta$?
3. In Table 6, why tuning AdamW eps? In general, is it suggested to use smaller eps for AdaGC?

**Limitations:**

yes

**Strengths And Weaknesses:**

Strengths:
1. This paper proposes AdaGC, an adaptive per-tensor gradient clipping scheme that mitigates such contamination by bounding gradient norms relative to a tensor-wise exponential moving average of their historical clipped values.
2. AdaGC is optimizer-agnostic, with minimal memory consumption and communication overhead, and good compatibility to hybrid parallelism.
3. Experiments show that AdaGC could eliminates training instabilities, and reduce loss spikes.

Weaknesses:
1. There is no corresponding theoretical analysis. It would be better if there is some theoretical guarantees of AdaGC + SGD, under some regular assumptions such as smoothness or convexity.
2. Tuning the extra hyperparameters ($\lambda$ and $\beta$) could make the training procedure more complicated and time-consuming. It would be better if there is some general guidelines or theories of how to choose these hyperparamers, instead of some empirical choice from grid search.

---

> ### Author Rebuttal · Authors · 2026-03-30
>
> We thank the reviewer for the positive evaluation and constructive questions.
>
> **W1: Theoretical analysis for AdaGC + SGD**
>
> We would like to point the reviewer to **Appendix F**, where we provide convergence analysis for Adam + AdaGC (Theorem F.1). Under standard assumptions **in non-convex settings**, we prove that Adam + AdaGC achieves **O(1/√T) convergence rate**, matching vanilla Adam. The proof techniques in Appendix F can be straightforwardly adapted to SGD + AdaGC, as the SGD case is strictly simpler (no moment estimation). We will add a brief summary of Theorem F.1 in the main text for better visibility.
>
> **W2: Hyperparameter selection guidance**
>
> Through grid search and cross-model validation, we recommend **λ_rel=1.04, β=0.99** as default values that generally work well without additional tuning. These defaults work well across all tested models: Llama-2 7B, Mixtral 8×1B, ERNIE 10B-A1.4B, and additionally **Qwen3 1.7B** in our new experiments (see response to Reviewer vxi5 for details). This is analogous to how GlobalGC's λ_abs=1.0 has become a standard default that practitioners rarely adjust.
>
> Moreover, AdaGC's relative threshold represents "how much deviation from history to tolerate"—a model-agnostic concept that is inherently more portable than GlobalGC's absolute threshold, which requires knowledge of expected gradient scale (model-dependent).
>
> We will add a "Practitioner's Guide" paragraph with these recommendations.
>
> **Q3: Why tune AdamW ε? Should smaller ε be used with AdaGC?**
>
> Recent works show smaller ε improves convergence:
> - Wortsman et al. (2023) "Small-scale proxies for large-scale Transformer training instabilities" uses ε=1e-15
> - LongCat-Flash Technical Report (2025) uses ε=1e-16
>
> Smaller ε allows more parameters to benefit from Adam's adaptive learning rates. However, smaller ε also makes training more spike-sensitive. As shown in Table 6: **GlobalGC + ε=1e-15 triggers spikes, while AdaGC + ε=1e-15 trains stably**. AdaGC enables using smaller ε for improved convergence while maintaining stability—a concrete practical benefit.

---

> > ### Author Rebuttal · Reviewer_TMiS · 2026-04-01
> >
> > The convergence guarantee of Adam + AdaGC is already available.
> > Although there is no theoretical analysis of how to tune the hyperparameters, the Practitioner's Guide is also acceptable.
> > I would recommend to add the discussion of "smaller ε also makes training more spike-sensitive" to the next revision.

---

### Official Review · Reviewer_vxi5 · 2026-03-13

**Soundness:** 2
**Presentation:** 3
**Significance:** 3
**Originality:** 3
**Overall Recommendation:** 4
**Confidence:** 5

**Summary:**

The paper proposes an adaptive tensor-wise clipping strategy to mitigate training instabilities. Extensive experiments across various settings, varying the architecture, optimiser, and other hyperparameters, are presented, demonstrating that AdaGC is stable during training compared with other clipping strategies.

**Compliance With Llm Reviewing Policy:**

Affirmed.

**Final Justification:**

Concerns are fully resolved.

**Key Questions For Authors:**

1. It is not clear why one should use adagc instead of qk-norm, or smaller beta_2 is not clear. I'm not trying to disregard the method because of this, but trying to understand where this tensorwise clipping might be beneficial compared to architectural changes.

**Limitations:**

yes

**Strengths And Weaknesses:**

## Strengths
1. The idea is simple, intuitive and optimiser and model agnostic.
2. Extensive experiments are conducted to show its merits.
3. Overall, the paper is well-written, the approach is clear, and the experiments are well-structured.

## Weaknesses
1. The main problem I see is that the paper tries to claim clipping is the best way to address training instabilities (line 135, column 2). This is arguable as there are effective techniques like QK-norm (see OLMO for a comprehensive analysis) and/or data-preprocessing to reduce instabilities. Also, compared to these approaches, clipping is not trying to address the cause but the effect -- which one cannot claim to be the best way. Note that edge-of-stability literature also tries to provide a theoretical understanding of this.
2. Related to the above, there needs to be experiments in realistic settings showing AdaGC is beneficial, on top of QK-norm and other stabilisation techniques (eg, smaller beta_2). This could be on a very large model or following [b] to induce instabilities at a small scale.
3. Zclip [a] is not discussed or compared. It seems very close to the proposed method, with a slight difference in how the EMAs are being aggregated and used.


[a] Kumar, Abhay, et al. "Zclip: Adaptive spike mitigation for llm pre-training." arXiv preprint arXiv:2504.02507 (2025).
[b] Wortsman, Mitchell, et al. "Small-scale proxies for large-scale transformer training instabilities." arXiv preprint arXiv:2309.14322 (2023).

---

> ### Author Rebuttal · Authors · 2026-03-30
>
> We sincerely thank the reviewer for the thorough and insightful feedback. We address each concern below.
>
> **W1: Positioning of clipping (line 135)**
>
> We agree our original claim was overly strong and will revise it. AdaGC is not meant to replace root-cause fixes like QK-norm or data preprocessing, but to serve as a **complementary, optimizer-level safety net**. The key advantage is **generality**: QK-norm addresses attention logit growth but not data corruption, hardware faults, or numerical precision issues; data preprocessing addresses data quality only. AdaGC operates at the gradient level, providing unified protection regardless of upstream cause—analogous to "defense in depth" in safety engineering.
>
> **W2 & W3: Experiments with QK-norm and comparison with ZClip**
>
> We conducted new experiments on **Qwen3 1.7B Dense** (with QK-norm enabled by default). The model is trained on 160B tokens from C4 with LR=4.5e-3 (known to induce spikes per Qiu et al. 2025 "Gated Attention"). Full results: https://anonymous.4open.science/r/AdaGC-ICML-Rebuttal-AC60
>
> | Method | Spike Score | #Spikes | 0-shot Avg | 2-shot Avg |
> |--------|-------------|---------|------------|------------|
> | GlobalGC | 0.2842% | 54 | 48.42% | 50.80% |
> | ZClip | 0.0421% | 8 | 48.42% | 51.41% |
> | **AdaGC** | **0.0053%** | **1** | **50.37%** | **52.64%** |
>
> **Key findings:**
> 1. **QK-norm alone is insufficient**: GlobalGC+QK-norm still produces 54 spikes.
> 2. **AdaGC provides significant value on top of QK-norm**: reduces to 1 spike, +1.95%/+1.84% accuracy improvement.
> 3. **AdaGC outperforms ZClip**: ZClip's global clipping reduces spikes to 8 but doesn't eliminate them, and provides no 0-shot improvement. AdaGC's per-tensor design precisely targets abnormal layers without over-constraining well-behaved ones (see Figure 6c: Global AdaGC allows 1 spike and 0.25% higher loss vs. per-tensor AdaGC).
>
> **Key Question: Why use AdaGC instead of QK-norm or smaller β₂?**
>
> They operate at different levels and are **complementary**, not mutually exclusive:
>
> | Approach | Level | Scope | Limitation |
> |----------|-------|-------|------------|
> | QK-norm | Architecture | Attention logit explosion | Attention layers only |
> | Smaller β₂ | Optimizer HP | Reduces second moment contamination | Changes optimizer dynamics |
> | AdaGC | Gradient | Wide range of gradient anomalies | Minimal per-tensor EMA overhead |
>
> In production LLM training, operators cannot always predict or prevent every instability source (data anomalies, hardware faults, numerical precision issues per our Table 1; optimizer hyperparameter sensitivity and lower-precision computation per our Figure 1). AdaGC provides general-purpose insurance against **a wide range of** abnormal gradient causes, which is why we recommend using it **alongside** architectural fixes.
>
> **ZClip discussion**: We will add ZClip to Related Work (Table 2). Key differences: (1) ZClip is global vs. AdaGC's per-tensor; (2) AdaGC validates on larger scale (10B, 1T tokens) and MoE architectures; (3) AdaGC has lower communication overhead (Section 4.3).

---

> > ### Author Rebuttal · Reviewer_vxi5 · 2026-04-01
> >
> > Thank you for providing results with QK-norm, comparing to zclip, and agreeing to correctly position clipping. Please include these results and writing changes in the final version. I'll increase the score.

---

### Decision · Program_Chairs · 2026-04-30

**Decision:**

Accept (regular)

**Comment:**

This paper proposes AdaGC, which will adaptively clip the per-tensor gradient to improve stability in LLM pretraining. Its main strengths are simplicity, optimizer agnostic, and promising results across settings. The main concerns raised by reviewers are regarding the comparison to other clip/stabilization methods, limited theoretical analysis, and hyperparameter sensitivity, but these were largely addressed in the rebuttal. Overall, the reviewers agreed that the paper is technically sound, well written and provides a useful practical contribution. Therefore, I recommend acceptance.